# Design and Evaluation of pH-Dependent Nanosystems Based on Cellulose Acetate Phthalate, Nanoparticles Loaded with Chlorhexidine for Periodontal Treatment

**DOI:** 10.3390/pharmaceutics11110604

**Published:** 2019-11-13

**Authors:** Gustavo Vidal-Romero, María L. Zambrano-Zaragoza, Lizbeth Martínez-Acevedo, Gerardo Leyva-Gómez, Susana E. Mendoza-Elvira, David Quintanar-Guerrero

**Affiliations:** 1Laboratorio de Posgrado en Tecnología Farmacéutica, Facultad de Estudios Superiores Cuautitlán, Universidad Nacional Autónoma de México, Estado de México C.P. 54745, Mexico; g.vidal23@hotmail.com (G.V.-R.); liz_martinez@comunidad.unam.mx (L.M.-A.); 2Laboratorio de Procesos de Transformación y Tecnologías Emergentes de Alimentos, Facultad de Estudios Superiores Cuautitlán, Universidad Nacional Autónoma de México, Estado de Mexico CP 54714, Mexico; luz.zambrano@unam.mx; 3Departamento de Farmacia, Facultad de Química, Universidad Nacional Autónoma de México, Ciudad de México 04510, Mexico; gerardoleyva@hotmail.com; 4Laboratorio de Virología, Facultad de Estudios Superiores Cuautitlán, Universidad Nacional Autónoma de México, Estado de México C.P. 54745, Mexico; seme@unam.mx

**Keywords:** nanocapsules, nanospheres, periodontitis, eugenol, buccal drug delivery, chlorhexidine base, cellulose acetate phthalate, pH-dependent polymeric

## Abstract

This work aimed to develop and evaluate pH-dependent systems based on nanospheres (NSphs) and nanocapsules (NCs) loaded with chlorhexidine (CHX) base as a novel formulation for the treatment of periodontal disease. Cellulose acetate phthalate (CAP) was employed as a pH-dependent polymeric material. The NSphs and NCs were prepared using the emulsion-diffusion technique and then characterized according to encapsulation efficiency (EE), size, zeta-potential, morphology, thermal properties, release profiles and a preliminary clinical panel test. The formulations showed 77% and 61% EE and 57% and 84% process efficiency (PE), respectively. Both systems were spherical with an average size of 250–300 nm. Differential scanning calorimetry (DSC) studies showed that the drug has the potential to be dispersed molecularly in the NSph matrix or dissolved in the oily center of the NCs. The CHX release test revealed that the release of NSphs-CHX follows Fickian diffusion involving diffusion-erosion processes. The NCs showed a slower release than the NSphs, following non-Fickian diffusion, which is indicative of anomalous transport. These nanosystems may, therefore, be employed as novel formulations for treating periodontal disease, due to (1) their coverage of a large surface area, (2) the controlled release of active substances at different pH, and (3) potential gingival tissue infiltration.

## 1. Introduction

Periodontal disease (PD) is one of the most important oral afflictions and is one that contributes to the global burden of chronic disease. Because PD is highly prevalent worldwide it represents a major public health problem in numerous countries. Like severe dental caries, PD is a major cause of tooth loss, which directly affects people’s quality of life by lowering their functional capacity and self-esteem and deteriorating social relationships [1]. Clinically speaking, PD is defined as an inflammatory condition of the gingiva and supporting tooth structures. Gingivitis and periodontitis are the most common forms [2].

The most common commercial products used to treat PD are oral rinses. However, not even extensive oral rinsing with antibacterial solutions reaches deep into subgingival tissues, while the high dosages of antibiotics required to achieve therapeutic levels in the periodontal pocket can result in unpleasant or even toxic adverse effects [3,4].

Various drug delivery systems for periodontal applications have been developed, including acrylic strips, fibres, films, injectable system gels, intra-pocket strips, vesicular systems, and microparticle and nanoparticle systems [5,6,7,8].

The drug candidate chosen for this study was chlorhexidine (CHX) base, which is widely used in clinical dental practice as an antiseptic oral rinse due to its activity against a wide range of microbial species. The CHX base (1,1’-hexamethylene-bis-5-(4-chlorophenyl) biguanide) is a symmetric molecule with two ionisable guanidine moieties in the form of a solid white crystal (m.p. 132 °C, M_W_ 505.5). The water solubility of the CHX base at 20 °C is 0.008% (*w/v*). The chemical structure of the CHX base is depicted in Figure 1. Its pKa values are 2.2 and 10.3, which render it dicationic over the entire range of physiological pH values [9,10,11]. This base is an antimicrobial agent often used in dentistry as an antiplaque agent but it has also demonstrated good activity against a wide range of oral bacteria. The CHX base is a drug of choice for treating periodontal disease because it prevents the development of an oral environment conducive to periodontal diseases [12,13].

Currently, 0.2% CHX aqueous solutions have been shown to be moderately effective in treating PD. Thus, there is growing interest in developing novel delivery systems that maintain the concentration of CHX for long periods to help reduce the incidence of these diseases [14]. Several nanocarriers or nanomaterials, such as liposomes, lipid, metal and polymeric nanoparticles, nanocrystals, dendrimers, and nanofibers, have been proposed as treatment options for PD [15]. They all use nanocarriers (e.g., organic nanoparticles) as drug delivery devices and have often been recommended as a means of meeting the specific requirements of the oral mucosa and enhancing conventional therapies by avoiding dilution effects while increasing the infiltration and adherence of the nanoformulations [11,16].

Cellulose derivatives (cellulose ethers and esters) have played a strong supporting role during the development of sustained, controlled-release oral dosages as (1) coatings capable of responding to changes in the physiological environment, (2) semi-permeable membranes, and (3) hydrophobic matrices that lower the dissolution rate of the active drug embedded in the matrix. Cellulose acetate phthalate (CAP) was one of the earliest and most effective polymers used for pH-controlled release. Its use continues today [17,18] because it resists prolonged contact with strongly acidic gastric fluid but dissolves in a mildly acidic or neutral intestinal environment [19]. This polymer does not form a gel in the presence of water but forms pH-sensitive and semi-permeable microporous systems. These properties make CAP a suitable excipient for developing new drug delivery systems that are now recognized as promising strategies for prolonging residence time and improving the specific localization of systems. Hence, they can reduce dosage frequency in controlled release formulations [20].

An alternative antibacterial approach involves antibiotic synergists. In antimicrobial therapy, synergism is used to describe the supra-additive activity of antibiotics when used in combination with other compounds. Eugenol (4-allyl-2-methoxyphenol) was selected as the model substance to form the oily core of the nanocapsule (NC) formulations prepared by the emulsification-diffusion method in order to potentiate the drug’s antimicrobial action and realize the synergistic, analgesic, and anti-inflammatory effect of the eugenol. This substance is the principle chemical component of clove oil derived from *E. aromatic* and has long been known for its analgesic, local anaesthetic, anti-inflammatory, and antibacterial effects [21,22,23,24,25,26]. Used in the form of a paste or mixture as a dental cement, filler, and restorative material, it belongs to the class of essential oils that are Generally Recognized as Safe (GRAS) by the U.S. Food and Drug Administration (FDA). Eugenol acts primarily by disrupting the cytoplasmic membrane [27,28] but its biological effects vary greatly depending on concentration. Eugenol may have a beneficial effect at concentrations ranging from 10^−8^ M to 10^−5^ M (prostaglandin synthesis, nerve activity, and white blood cell chemotaxis inhibition) but can be cytotoxic at concentrations >10^−3^ M (cell death, cell growth, and respiratory inhibition) [29]. Today, the development of innovative drug delivery systems focuses on designing multiple options, blocking the adverse effects of drugs, and reducing dosage intervals.

Oral fluid can be considered the protective medium for all tissues of the oral cavity. It acts as a buffer by maintaining a pH between 5.75 and 7.05. This fluid is composed mainly of water (99.5%), organic compounds (0.3%), and inorganic and trace elements (0.2%) [30]. By contrast, the periodontal microenvironment is more anaerobic and perfused by a plasma filtrate called gingival crevice fluid. The growth of microorganisms in this microenvironment has been shown to cause periodontal disease. Because the site of bacterial infection is usually inaccessible to agents present in the oral cavity, antimicrobial agents administered there tend to be ineffective. Therefore, a local antibiotic agent loaded in an intra-pocket delivery system promotes a high drug concentration in the gingival crevice fluid that reduces adverse effects and provides such advantages as improving drug efficacy and patient compliance [31].

Nanometer-sized systems have the potential to infiltrate easily into the periodontal pocket and remain in the damaged tissue where they release the drug in a controlled manner [32]. PDs usually involve diverse, complex mechanisms that require various molecules to achieve effective treatments. Because the bacteria that cause periodontal tissue inflammation and destruction are present in every stage of PD, treatment requires antimicrobials, antioxidants, antiresorptives, and anti-inflammatory drugs. At present, some commercial products for treating PD include mouthwashes that are applied daily, systemic antibiotics, and systems for the local delivery of bioactive agents [33].

The aim of the present study was to develop two nanoparticle systems—nanospheres and nanocapsules—loaded with CHX base as novel formulations to improve treatment of PD by (1) enabling the maintenance of therapeutic drug levels over long periods, (2) preventing relapses, (3) decreasing the amount of drug needed to achieve the therapeutic effect, and (4) reducing treatment time. These factors should also result in greater patient acceptance.

## 2. Materials and Methods

### 2.1. Materials

Cellulose acetate phthalate (MW 2534.1) was purchased from Vita Drug, Mexico. Eugenol oil (≥98%) was supplied by Dentalflux, Mexico. Poly(vinyl alcohol, PVA: Mowiol 4-88, molecular weight ~31,000) and chlorhexidine base were obtained from Sigma Aldrich (St Louis, MO, USA). HPLC-grade methyl ethyl ketone, methanol, ethanol, and acetonitrile were purchased from Productos Quimicos Monterrey, S.A. de C.V. (Nuevo León, Mexico). Distilled water was obtained from a RiOs^TM^ distiller (EMD, Millipore^®^, Billerica, MA, USA). All other reagents were of at least analytical grade and used without further purification.

### 2.2. Preparation of Nanospheres and Nanocapsules by the Emulsification Diffusion Technique

The CAP nanospheres and nanocapsules loaded or unloaded with CHX (CHX-CAP-NSphs/CHX-CAP-NCs) were prepared using an emulsification-diffusion technique adapted from one described previously [11,34]. Briefly, the organic solvent (methyl ethyl ketone) and water were mutually saturated for at least 20 min before use to ensure the initial mass equilibrium of both liquids. For the NC formulations, 200 mg of the pH-dependent polymer (CAP), 365 mg of eugenol oil, and the amount of CHX indicated for each formulation were dissolved in 20 mL of water-saturated organic solvent. This organic solution was emulsified with 40 mL of organic solvent-saturated aqueous solution at 5% (*w*/*v*) of PVA using a stirrer (Caframo RZR-1, Staufen, Germany) at 1700 rpm for 10 min. The resulting mixed phase immediately turned milky with a bluish opalescence (i.e., the Tyndall effect) due to the instantaneous formation of nanocapsules, according to the formulation (Table 1). The excess of organic solvent was eliminated from the raw nanocapsule suspensions by vacuum steam distillation at 30 °C and 30 rpm using a Laborata 4000 efficient (Heidolph^®^ Instruments GMBH &Co., Schwabach, Germany). Next, the nanocapsules were isolated by centrifugation (Optima^®^ LE-80, Beckman Coulter, Inc., Fullerton, CA, USA) at 12,000 rpm and 15 °C for 40 min, followed by three washes with distilled water under the same conditions to remove the excess PVA adsorbed on the nanosystems’ surface. Finally, the systems were resuspended in a minimal volume of distilled water. This optimized procedure and formulation were also used to obtain the nanospheres (NSphs), though this operation does not require including eugenol in the formulation.

### 2.3. Physicochemical Characterization of the Cellulose Acetate Phthalate (CAP)-Nanospheres (NSphs)/CAP-Nanocapsules (NCs)

#### 2.3.1. Particle Size Measurement and Surface Charge

The particle size of each prepared sample was analysed with a Dynamic Light-Scattering System (Zetasizer^®^ Nano ZS, Malvern Instruments Ltd., Worcestershire, UK). Samples of the nanosystems were suspended, diluted appropriately with Milli-Q^®^ quality ultrapure water (Millipore Corporation, Bedford, MA, USA) to achieve the appropriate particle concentration for measurement (*n* = 3). Analysis was performed at a scattering angle of 90° for 180 s at 25 °C. Z-potential was also determined with the aforementioned Zetasizer^®^ Nano ZS but by utilising a capillary bending cell with 150 V of electric current and after diluting the samples with deionized water. Measurements were made in triplicate at 25 °C.

#### 2.3.2. Scanning Electron Microscopy

SEM was used to examine the surface morphology of the CAP NSphs and NCs loaded with CHX (CHX-CAP-NSphs/CHX-CAP-NCs). A droplet of each sample was spread on a glass surface, left to dry, and then mounted on stubs and shadowed in a cathodic evaporator with a gold layer (~20 nm) using a JFG-1100 Sputter Coater (JEOL, Tokyo, Japan). The samples were observed under a scanning electron microscope (LV-SEM JSM-5600, JEOL, Tokyo, Japan) at 15–25 kV electron acceleration voltage and a pressure of 12–20 Pa in the specimen chamber. To evaluate the potential infiltration of the nanoparticles into the gingival pocket, unwaxed nylon-dental floss (Superfloss^TM^ Oral-B^®^ Belmont, CA, USA) was immersed in a 5% (*w*/*v*) suspension of CHX-CAP-NSphs/CHX-CAP-NCs, dried at room temperature, and characterized by SEM.

#### 2.3.3. Stabilizer Quantification

The residual amount of PVA on the nanoparticles was determined in triplicate by a colorimetric method. Considering that PVA in solution forms stable complexes with iodine in the presence of boric acid, the method employed was as follows: approximately 5–10 mg of the nanosystems (CHX-CAP-NSphs/CHX-CAP-NCs) were digested in 5 mL of methyl ethyl ketone for 48 h. The solvent was then evaporated, the suspension was filtered in a 0.22 μm Millipore^®^ mesh, and the volume was adjusted to 10 mL with water (Solution A). Next, 2 mL of a 0.65-M boric acid solution and 1 mL of an iodine solution (0.05 M iodine and 0.15 M potassium iodine in water) were added to 5 mL of Solution A. The absorbance of the resulting samples was measured at 640 nm in a Varian Cary^®^ 50 UV-vis spectrophotometer (Walnut Creek, CA, USA) using as the target a solution of 5 mL of water with 2 mL of boric acid solution (0.65 M) and 1 mL of iodine solution (0.05 M iodine and 0.15 M potassium iodine) [35]. The calibration curve for PVA quantification was linear over the 5–50 μg/mL range (*r^2^* = 0.9999).

#### 2.3.4. Determination of Encapsulation Efficiency and Chlorhexidine Loading Capacity

The entrapment efficiency (EE) refers to the amount of drug that can being entrapped or encapsulated with respect to the initial amount of drug in the formulation (Equation (1)). The loading capacity refers to the amount of drug that is contained in the nanosystems with respect to the total amount of nanosystems obtained (Equation (2)). Process efficiency (PE) refers to the amount of nanosystems (NCs or NSphs) that can be obtained with respect to the initial amount of materials present in the formulation (Equation (3)). Equations (1)–(3) were used to evaluate the EE, drug-loading capacity, and PE.

(1)$$Entrapment\;or\;encapsulation\;efficiency=\frac{Amount\;of\;drug\;loaded}{Initial\;amount\;of\;drug}\times 100$$

(2)$$Loading\;capacity=\frac{Amount\;of\;drug\;in\;nanosystems}{Total\;amount\;of\;nanosystems\;obtained}\times 100$$

(3)$$Process\;efficiency=\frac{Total\;amount\;of\;nanosystems\;obtained}{Initial\;amount\;of\;materials\;in\;formulation}\times 100$$

The NCs were prepared with different amounts of the active ingredient (10, 20, and 30 mg) but the same amount of eugenol oil: 365 mg (Table 1). The CHX content in the CHX-CAP-NCs was then determined. Briefly, 10 mg of CHX-CAP-NC powder was dissolved in 5 mL of methyl ethyl ketone and 4 mL of NaOH [0.1M] was added to achieve the total dissolution of the CAP and release the encapsulated CHX. The organic and aqueous phases were separated and dried at room temperature and the solid residue was re-suspended in 10 mL of methanol. A sample of this solution was filtered in a 0.22-μm Millipore^®^ mesh and analysed by HPLC to determine the amount of encapsulated drug. The same procedure was employed for the CHX-CAP-NSphs according to the best formulation.

#### 2.3.5. Quantitative Determination of the Chlorhexidine Base

The CHX base was quantified by reversed phase adsorption chromatography using a LiChropher^®^ 100 RP-18, 5 μm (125 mm × 4 mm) HPLC cartridge (Merck, Darmstad, Germany) on a Varian ProStar HPLC system (Chromatography Systems, San, CA, USA). This device was equipped with a pump (model PS 210), autosampler (model PS 400), and multi-wavelength UV-visible detector (model PS 320) connected to a PC interface. Elution was carried out under isocratic conditions using a mixture of acetonitrile and 30 mM of sodium acetate buffer as the mobile phase (35:65) adjusted to pH 3.3 with acetic acid (96%) at 0.5% of triethylamine. The sample volume injected was 20 μL. Detection was performed at 260 nm and a flow rate of 1 mL/min. The linear calibration curve for CHX was established in a range of 10–80 μg/mL. The retention time for the CHX was approximately 3.5 min. Chromatograms were analysed with a Galaxy Software package following a method adapted from H. Lboutounne et al. [11].

#### 2.3.6. Differential Scanning Calorimetry (DSC)

DSC was used to characterize the thermal properties of the components in the optimized formulations, i.e., unloaded CAP-NCs and CHX-CAP-NCs. The samples were weighed (3–5 mg) directly in hermetic aluminium pans and scanned in a temperature range of 0–400 °C at a heating rate of 10 °C/min under a nitrogen flux of 50 mL/min utilizing a previously calibrated and adjusted calorimeter (DSC Q10, TA Instruments, New Castle, DE, USA).

### 2.4. In Vitro Drug Release Study

The method used was adapted from the one described by Silvana Gjoseva et al. [36]. Briefly, drug release studies were performed under sink conditions using an amount equivalent to 1 mg of CHX for each dry nanosystem (NSph or NCs). This was suspended in 20 mL of phosphate buffer solution at pH 7 and 2.5% Brij^®^58 (SBF pH 7) and then placed in closed thermostatic test tubes at 37 °C using an agitated water bath at 50 rpm. The dissolution conditions simulated the pH and ionic strength of saliva/gingival fluid (pH 7 and 75 mmol/L, respectively). At various predetermined time intervals (NSphs, 0–20 min; NCs, 0–100 min), a 2 mL aliquot of the release medium was removed and replaced with fresh SBF at pH 7. The amount of CHX released was determined using the HPLC method described above. The data obtained were fitted using Higuchi and Korsmeyer-Peppas models to ascertain the transport mechanism, release type, and drug release kinetics.

### 2.5. Statistical Analyses

An analysis of variance (ANOVA) with between-means comparisons using a t-students test at a significance level of 0.05 was used to evaluate the influence of the amount of drug on particle size, Z-potential, encapsulation efficiency, and process efficiency. Another analysis of variance was performed using STATGRAPHICS^®^ Centurion XVI software 16^th^ (Statpoint Technologies, INC).

### 2.6. Panel Test

Healthy volunteers with periodontal disease were diagnosed and invited to participate in the study. Research was conducted following the principles on experimentation involving human subjects outlined in the Helsinki Declaration. This project was registered with the Postgraduate and Investigation Division of the Facultad de Estudios Superiores Zaragoza UNAM (approval number FESZ-RP/17-118-005) and accepted by Investigation, Ethics, and Biosafety Committees of health center TII Manuel Gutiérrez Zavala of Mexico City (ID MGZ-2017-085, 16 September 2017) A panel test (*n* = 6) was evaluated using CHX-CAP-NCs versus a group treated with an approach that included a mouthwash (*n* = 6). The procedure was explained to the participants prior to performance. After receiving their authorization in the form of a signed informed consent report in which they agreed to follow the treatment indications, an odontoxesis process was carried out that involved periodontal probing and removal of the supragingival calculus before the O’Leary index was applied to determine the presence of dentobacterial plaque (PDB). To evaluate the index, the patient received a revealing pill which dissolved in the mouth and stained surfaces containing PDB, as a result of which areas were visually pigmented and recorded on a registration form, where each tooth was divided into four sectors (mesial, vestibular, distal, and lingual faces). To determine the final score (average), the total number of faces with plaque was added and this number divided by the number of faces present in the mouth and multiplied by 100; this score can be compared [37,38,39]. This procedure was finalized with a prophylaxis treatment, after which a dispersion of CHX-CAP-NCs in amounts equivalent to 9 mg of CHX was applied every third day for 15 days in the form of soft acetate protectors placed on the patients’ gums. System effectiveness was monitored by the control of microbial plaques using the O’Leary index as an indicator of the reduction of microbial plaque and for comparison with the untreated group.

## 3. Results and Discussion

### 3.1. Physicochemical Characterization of the CAP-NSphs/CAP-NCs

Table 2 shows the particle size of the nanosystems. All batches had sizes below 500 nm. It is important to emphasize that the CAP nanocapsules functionalized with eugenol and CHX had sizes 60% and 79% greater than the control nanospheres, while the nanospheres loaded with CHX-CAP showed an increase of only 37% with respect to the controls. This indicates that 33–42% of the increase in particle size is attributable to the presence of eugenol oil, which was incorporated into the nanocapsule formation by the emulsification diffusion method due to its synergetic effect with CHX [40,41]. Piñon Segundo et al. (2005) found that nanoencapsulation of triclosan using CAP generated sizes of 192–235 nm, which increased with higher amounts of triclosan [32]. We hypothesize that the increase in particle size can also be attributed to the presence of eugenol oil in the nanocapsule entities and the molecule size of CHX. In addition, the presence of OH- groups allows the system to stabilize by forming bound hemiacetals [42]. In the NCs, the change in the CHX concentration from 10 to 20 mg increased particle size and slightly increased the polydispersity index at 30 mg. The differences between the NSphs and NCs may be attributable to the effect of the oil on the interfacial behaviour during the formation process (diffusion-stranding mechanism), since this increased the aggregation of the proto-nanoparticles due to a larger diffusion layer. When the amount of CHX in the formulation increased, the particle size also increased slightly, as did the polydispersity index, indicating that the amount of drug (10–30mg) did not have a significant effect on nanocapsule size (*p* value 0.4797).

The Z-potential measures the degree of repulsion between adjacent, similarly charged particles in a dispersion. In general, when this potential is relatively low, attraction exceeds repulsion and the dispersion will flocculate [43]. The −20 mV (Table 2) Z-potential of all our CHX-loaded batches suggests that the nanosystems are stable in dispersion due to their negatively charged surfaces. The residual amount of PVA was less than 5% in all batches. This amount suffices to give the nanocapsule dispersion good stability. It is well-known that this PVA concentration adsorbed on the surface of nanoparticles prevents aggregation by steric repulsion as it forms a stable, strongly-attached thick layer [31,34]. The CHX load, by contrast, has an electric charge that contributes to increasing the Z-potential by approximately 10 mV. This behaviour may indicate that the drug is entrapped inside the nanoparticles.

The spherical shape of the drug-loaded NCs and NSphs was confirmed by SEM (Figure 2a,b). SEM micrographs confirm the sub-micronic size and show that both nanosystems have smooth homogeneous surfaces with no evidence of CHX crystals.

To show the infiltration capability of these nanosystems, the NSphs and NCs were infiltrated in dental floss by immersion in a 5% (*w*/*v*) suspension of each nanosystem. After immersion for 24 h, the amount absorbed of nanosystems corresponded to 5.8 mg for NCs and 2.8 mg for NSphs. Figure 2c,d illustrate that both nanosystems were retained in the small spaces of the threads of the dental floss, a finding indicative of their potential to infiltrate and be absorbed on surfaces. More specifically, the small size of these systems allowed them to infiltrate the periodontal pocket when applied in suspension, so flosses loaded with nanoparticles could be another means of administering CHX-CAP-NCs.

Table 2 shows that process efficiency varied with the amount of CHX in the range 69–84%. A high amount of the active ingredient decreased process efficiency because more CHX increased aggregation of the polymer chains. Also, it seems that CHX decreased stabilizer efficiency [44]. For all batches of NCs prepared with different amounts of drug, results show a statistically significant difference in process efficiency at a significance level of 95 (*p* value = 0.01). Possibly, the efficiency of process in batches of NCs loaded with 10 mg of CHX is greater than batches of NSphs with the same amount of CHX due to the presence of eugenol oil where the CHX was dissolved prior to addition in polymer solution, so the aggregation of the polymer chains is much smaller, giving greater efficiency in the process (84.30%). It has been proposed that the presence of eugenol promotes the formation of a defined interface of nanoil/medium (nanoemulsion) where the polymer coacervate forms a thin film around it. By contrast, NSph formation depends on the polymer aggregation and functionality of the stabilizer to prevent the protonanoparticles coalescence which is more influenced by the CHX presence (process efficiency 57.17%) [45]. The encapsulation efficiency of the CHX in the CAP-NCs was approximately 60%. This behavior can be explained because during its formation process it was not possible to encapsulate the total amount of oil present in the formulation, meaning that part of the drug dissolved in the non-encapsulated oil was lost. In these batches the loss of drug into the continuous phase is constant to the conditions of work. On the other hand, in the formation of NSphs it is possible to obtain high entrapment efficiencies (77%) because the drug is mixed directly with the polymer matrix and only a small quantity of CHX is not entrapped into the polymer chains. Similar results have been reported by Lboutounne et al., who encapsulated a CHX base in polymeric nanocapsules with poly(ε-caprolactone) as the polymer and Labrafac hydrophile^®^ WL 1219 as the oil [11].

It has been noted that it is difficult to achieve high encapsulation and process efficiency because the process involved requires several purification stages that could allow drug release and loss before nanocapsule formation. In this regard, and considering the volatility of eugenol, the final amount of this oil that remained in the CHX-CAP-NCs post-processing was determined by the HPLC method described above, which found a loss of approximately 90% with respect to the initial amount. Ying Shao et al. have reported an EE of eugenol in eugenol-chitosan nanoemulsions of around 11.61% when prepared by ultrasonic treatment employing a chitosan-acetic acid solution containing 1 *wt*/*v*% Tween 20 at ratios of chitosan-to-eugenol of 1:1 [23]. This quantity, however, is considered sufficient to contribute to the therapeutic effect, since concentrations of 10^−8^ to 10^−5^ M have been reported as effective in periodontal treatment [29,46]. It is important to point out that when combined with some antibiotics eugenol has a synergistic impact that increases the antibiotic’s therapeutic effect [27,47]. On this topic, Hanene Miladi et al. have reported a synergistic effect of eugenol oil and tetracycline with a reduction rate ranging from two-to-eight-fold in the minimal inhibitory concentration [48].

The loading capacities for the batches of NCs were 2.72% for the lowest (10 mg), 5.44% for the medium (20 mg), and 7.65% for the highest concentration (30 mg). Thus, the CHX in CAP-NCs does not show a statistically significant influence on size, though it does determine loading capacity and process efficiency.

DSC studies were performed to understand the relations among the ingredients of the formulations. Thermograms of free CHX as well as the polymer, stabilizer, unloaded NCs, loaded NCs, and the physical mixture were obtained to define the physical state of the drug and the polymer in the NCs, and to detect drug-polymer interactions. NCs were used as a model system because their elaboration requires most of the components used in nanospheres. Figure 3 shows the melting endotherm of the pure CHX base at 135.65 °C, followed by its degradation. The physical mixture shows a broad peak between the melting temperatures of eugenol oil and CHX corresponding to the interaction of two substances. The decrease in the endothermic peak of CHX base in this physical mixture could be attributed to a dilution effect. In addition, Figure 3e shows slight endothermic changes that indicate the presence of PVA and CAP. In the case of the CHX incorporated into the CAP-NCs (Figure 3f), the endothermic peak characteristic of the melting point of CHX was not observed at all concentrations assayed. This behavior suggests that CHX dissolves into the eugenol encapsulated in a polymeric membrane composed of the polymer. One particularly important observation is the absence of the crystalline peak of the CAP in the CHX-CAP-NCs, which suggests that CHX and eugenol may act as plasticizers of the polymer.

### 3.2. In Vitro Drug Release Study

Figure 4 and Figure 5 show the release profiles of CHX for the CHX-CAP-NSphs and CHX-CAP-NCs, respectively. A rapid release of CHX is evidenced for both systems. Complete release was obtained within 15 min for the CHX-CAP-NSphs and 80 min for the CHX-CAP-NCs at pH 7 in a phosphate buffer solution with Brij^®^ 58 2.5% added. This difference can be attributed to the architecture of the nanosystems, as NSphs have a matrix structure while the structure of NCs is capsular. A matrix system usually releases through diffusion, whereas a capsular system can release by either diffusion through the membrane or breaking the capsule. It is to be expected that similar behavior may occur under in vivo conditions. This drug is in class II of the biopharmaceutics drug classification, which means that dissolution is the limiting factor, as we simulated in the in vitro test. Class II drugs have an expected in vitro–in vivo correlation that allows a good prediction of formulation performance in vivo based on in vitro drug release profile studies [49,50].

Conventional pharmaceutical forms used to treat periodontal disease generally require several applications throughout the day due to the rapid elimination of the drug from the site. Indeed, only shorter intervals between administrations can ensure a local pharmacological effect [51], but these additional applications come at the cost of the patient’s comfort and can affect treatment compliance. The systems proposed herein can improve retention in the periodontal pocket, meaning fewer administrations and a high therapeutic effect can be projected. Because CHX-CAP-NSphs and CHX-CAP-NCs are pH-dependent systems, rapid drug release is expected in cases of chronic periodontal disease because the pH of the crevice fluid promotes the rapid dissolution of the CAP polymer. This occurs because the environment tends to become acidic during infection due to the combined actions of bacterial metabolism and the host’s immune response [14,52].

The results of the mathematical modelling of the in vitro release results are shown in Table 3. To determine whether CHX transport is based on Fickian or non-Fickian diffusion, the Korsmeyer-Peppas and Higuchi models were used. In both cases—i.e., nanospheres and nanocapsules—the Korsmeyer-Peppas model presented a satisfactory fit. This semi-empirical model was developed explicitly for polymeric matrices and is used to elucidate the release mechanism because of its ability to differentiate between, and then categorize, distinct geometrical systems by interpreting their exponents (*n*). In the present case, the geometrical systems had a spherical tendency, which theoretically means values for an *n* = 0.5 slab for Fickian diffusion and higher values of *n*, i.e., between 0.5 and 1, or *n* = 1, for mass transfer under non-Fickian diffusion [53,54].

The CHX-CAP-NSphs showed a rapid release of the total amount of CHX at 15 min. This adjusted to the Korsmeyer-Peppas model by showing a determination coefficient of 0.913, a diffusion exponent (*n*) equal to 0.471 and *K*_H_ = 0.2402 mg/min^1/2^. These results indicate that the release of CHX followed Fickian diffusion that involved diffusion-erosion processes due to CHX’s molecular diffusion from the matrix of NSphs as a consequence of its chemical gradient and relaxation of CAP’s polymer chains. The CHX-CAP-NCs had a slower release than the NSphs as they had liberated the total amount of drug at 80 min, with a diffusion exponent of *n* = 0.6083 and a delivery release of *K*_H_ = 0.1313 mg/min^1/2^. Thus, the NCs had non-Fickian diffusion; that is, an anomalous transport. We hypothesize that this behavior may occur because the eugenol oil acts as a plasticizing agent of the CAP polymer that likely delays drug release in the pH = 7 Brig^®^58 2.5% buffer solution. This would also relax the CAP’s polymer chains that form the nanocapsule membrane to allow drug diffusion towards the medium to a point where this becomes completely eroded and the total amount of the drug can be released.

Comparing the results obtained from both nanosystems allows us to affirm that Fickian diffusion would provide the most predominant release mechanism, though it is important to note that changes in the pH of the release medium can delay the release time of the total amount of drug. This property could permit prolonged release over long periods because the pH of the crevicular fluid in the periodontal pocket changes from around 5 in normal conditions to 7 as PD progresses due to inflammation and the accumulation of bacteria. The NCs that contained eugenol oil probably acted first to reduce tissue inflammation and the pH of the medium thanks to their anti-inflammatory and antimicrobial properties. As the pH of the medium (i.e., the crevicular fluid in the periodontal pocket) decreased, the nanocapsules are able to lower their release rate because the CAP polymer becomes less soluble at acid pH. The decrease in the pH of the medium results from an improvement in the stage of PD. In the case of severe and chronic periodontal disease, the pH of the periodontal pocket that contains the crevicular fluid is slightly basic, i.e., close to 7, but as the presence of bacteria decreases the stage of the disease improves until the inflammation and other characteristic discomforts disappear. It is at that point that the pH of the periodontal pocket changes from slightly basic to slightly acidic, so we would expect that this decrease in the pH of the periodontal pocket medium, where the NSphs and NCs infiltrated and are deposited, will retard drug release from nanosystems formulated with the CAP polymer. The nanoparticle infiltration has been documented by our group (Ganem-Quintanar, 1997) using confocal laser scanning microscopy. Thus, when NPs are gently applied to the porcine gingival sucular space they are able to penetrate into the junctional epithelium. These results suggest that NPs can provide a potential intrapocket carrier system for delivery of active substances to the periodontal pocket [55].

### 3.3. Panel Test

Figure 6 presents the clinical evolution of patients with periodontitis, which is characterized by inflammation of the gums and accumulation of the supragingival calculus even with no movement of the dental pieces. The patients were treated with the procedure outlined above, receiving instructions to administer the equivalent of 9 mg of the CHX contained in the NC dispersion every third day for 15 days. The treatment was applied to buccal protectors that were used only at night, when bacterial activity is highest. The O’Leary index of dentobacterial activity showed a decrease in plaque, as seen in in Figure 6, which ranged from 76–26% using CHX-CAP-NCs and 91–67.5% using a commercial product (mouthwash). This finding is corroborated by the patients that show a gradual decrease in gum inflammation. It was also possible to confirm that patient compliance was acceptable, since this treatment does not require several applications per day, but only one nocturnal application. Thus, patients can carry out their normal daily activities without the bother of a protector. Finally, these systems have a soft consistency that does not irritate the oral cavity. It is important to mention also that the dose administered is lower than the one usually applied in commercial formulations. Rodrigues et. al. have evaluated the clinical response of periodontal tissue to the application of PerioChip compared to scaling and root planning (SRP) and identified a greater reduction in pocket depth and clinical attachment gain when local drugs were used in deep pockets [56]. Dorota et. al. have demonstrated the decrease in periodontal pocket depth and bleeding after clinical application of hydroxyethyl cellulose matrices loaded with metronidazole [57]. A study carried out by Norma et. al. showed a 26% reduction in dentobacterial plaque after a 14 day treatment using a 0.12% CHX mouthwash [58].

## 4. Conclusions

The emulsification-diffusion technique used in this work allowed the preparation of two systems—CHX-CAP-NSphs and CHX-CAP-NCs—that can be applied as local drug delivery systems in the oral cavity. More specifically, they can infiltrate, or be administered into, the periodontal pocket. The NCs showed the best entrapment efficiency at 84.30%. DSC showed that introducing CHX into CHX-CAP-NC systems decreases the glass transition temperature of the polymer. The CHX release test showed that the CHX-CAP-NSphs and CHX-CAP-NCs adjusted to the Higuchi and Korsmeyer-Peppas models and so correspond to Fickian diffusion that is dependent on time in the case of NSphs, and non-Fickian diffusion (anomalous transport) in that of the NCs, perhaps because eugenol oil acts as a polymer plasticizer. Both processes involve diffusion-erosion processes. These results indicate that CHX-CAP-NSphs and CHX-CAP-NCs could be of great help in treating periodontal disease because they affect a larger surface area and have both controlled release and potential tissue infiltration. In addition, eugenol could potentiate the antibacterial effect of CHX and exert a synergistic effect that takes advantage of its analgesic and anti-inflammatory properties.

Applying CHX-CAP-NCs reduced the dentobacterial plaque index by 65.78%, in contrast to a commercial mouthwash, which reduced this index by only 25.8%. Therefore, we can suggest that nanoparticulate systems allow much greater infiltration into the subgingival tissue (periodontal pocket) because they cover a greater surface area and so reduce both the therapeutic dosage and treatment time.

## Figures and Tables

**Figure 1 pharmaceutics-11-00604-f001:**
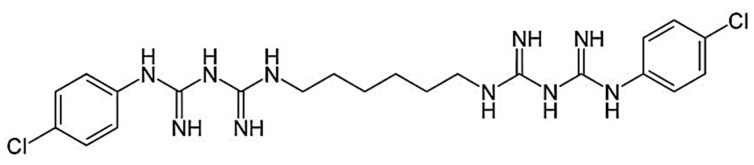
Chemical structure of the chlorhexidine (CHX) base.

**Figure 2 pharmaceutics-11-00604-f002:**
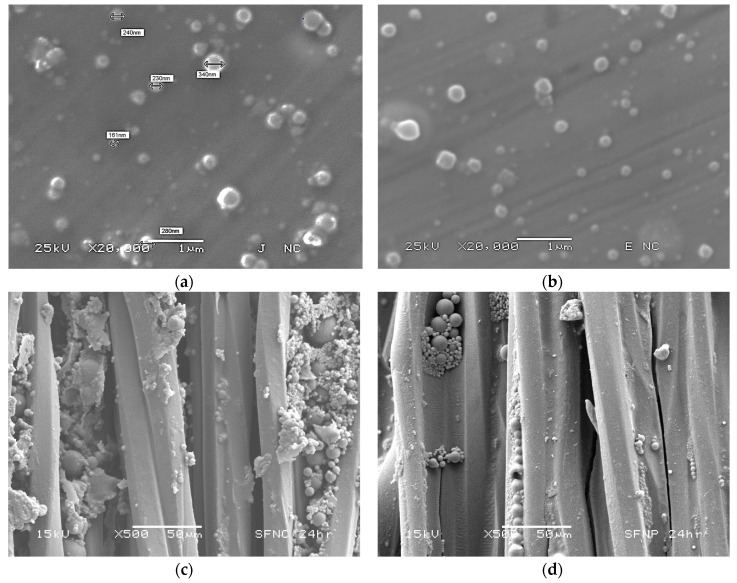
(**a**) Micrographs of CHX-CAP-NCs; (**b**) CHX-CAP-NSphs; and (**c**) CHX-CAP-NCs adsorbed on the surface of dental floss, and (**d**) CHX-CAP-NSphs adsorbed on the dental floss surface.

**Figure 3 pharmaceutics-11-00604-f003:**
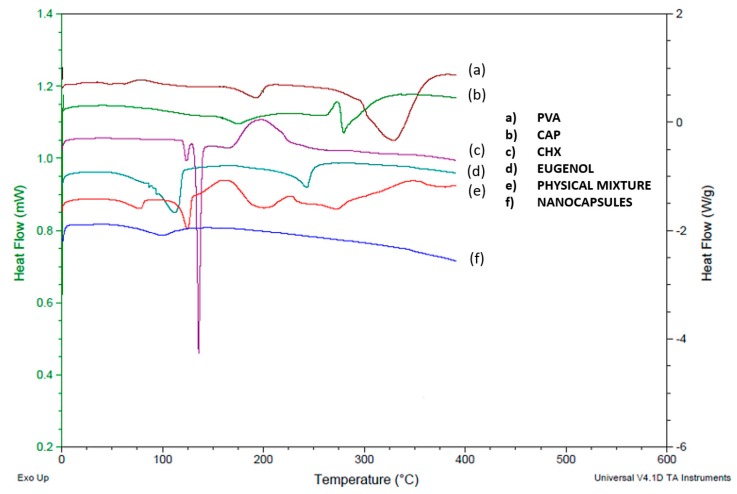
Thermograms of individual excipients, the physical mixture, and CHX-CAP-NCs.

**Figure 4 pharmaceutics-11-00604-f004:**
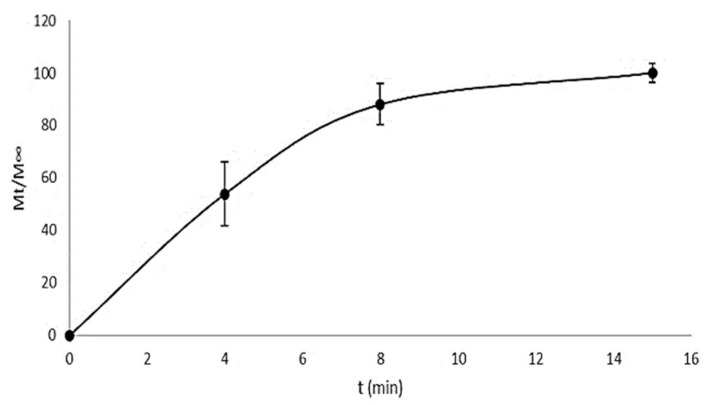
Release profile of CHX from NSphs.

**Figure 5 pharmaceutics-11-00604-f005:**
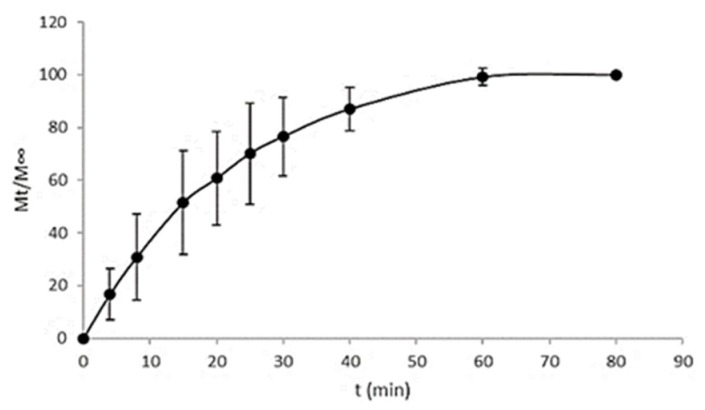
Release profile of CHX from NCs.

**Figure 6 pharmaceutics-11-00604-f006:**
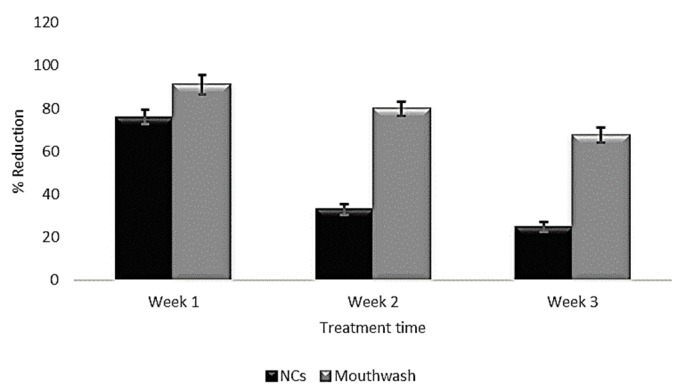
Change in dentobacterial plaque of CHX CAP-NCs versus a commercial mouthwash.

**Table 1 pharmaceutics-11-00604-t001:** Proposed formulations for preparing chlorhexidine cellulose acetate phthalate nanocapsules (CHX-CAP-NCs) and CHX CAP nanospheres (CHX-CAP-NSphs). Legend: PVA, polyvinyl alcohol.

Material	NCs			NSphs
	Formulation A	Formulation B	Formulation C	Formulation
Polymer (CAP) (mg)	200	200	200	200
Oil (eugenol) (mg)	365	365	365	– *
Drug (CHX) (mg)	10	20	30	10
External phase solvent (PVA 5%) (mL)	40	40	40	40
Inners phase solvent (MEC **) (mL)	20	20	20	20

* For this formulation eugenol oil is not necessary. ** MEC (Methyl Ethyl Ketone).

**Table 2 pharmaceutics-11-00604-t002:** Results of the physicochemical characterization of the control, CHX-CAP-NC, and CHX-CAP-NSph batches.

Formulation	Amount of CHX (mg)	Size ± SD (nm)	PI * ± SD	Z-Potential ± SD (mV)	Process Efficiency ± SD (%)	Entrapment Efficiency ± SD (%)
Control	–	180.6 ± 0.70	–	−10.83 ± 0.77	–	–
NCs	10	290.65 ± 15.70	0.14 ± 0.02	−20.16 ± 2.64	84.30 ± 0.74	61.93 ± 4.28
	20	324.46 ± 55.46	0.238 ± 0.02	−18.77 ± 3.46	72.00 ± 1.66	64.49 ± 0.80
	30	296.35 ± 39.00	0.291 ± 0.05	−18.71 ± 5.21	69.25 ± 0.43	59.88 ± 2.77
NSphs	10	247.60 ± 9.61	0.242 ± 0.01	−20.35 ± 1.91	57.17 ± 1.23	77.36 ± 0.62

* PI (Polydispersity Index).

**Table 3 pharmaceutics-11-00604-t003:** Correlation coefficients and constants of the kinetic models, Peppas equation, and Higuchi model used to determine the transport mechanism, release type, and kinetics of drug release from nanospheres and nanocapsules.

Nanosystem	Korsmeyer-Peppas Model		Higuchi Model	
	*r* ^2^	*n*	*r* ^2^	K_H_ (mg/min^1/2^)
NSphs	0.913	0.471	0.890	0.2402
NCs	0.9464	0.6083	0.9343	0.1313

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
