# Peer review of "Design and Evaluation of pH-Dependent Nanosystems Based on Cellulose Acetate Phthalate, Nanoparticles Loaded with Chlorhexidine for Periodontal Treatment"

_pharmaceutics, 2019, doi:10.3390/pharmaceutics11110604_

Round 1

Reviewer 1 Report

Paper described polymeric nanospheres and nanocapsules of chlorhexedine base for periodontal treatment. Please see following comments: 

Please correct "Ph" in title to pH Please remove claims on gingival tissue infiltration from abstract as they are unsupported.  Introduction: Please clearly state novelty of proposed formulations compared to previously developed and cited formulations.  Eugenol activities are summarised in intro. Please support with appropriate evidence (references) Materials: Please state purity of Eugenol Particle size method: Were particles diluted for sizing? If so in which media and what dilution was used.  Please include release studies in pH 5 and pH7. Agree that release is likely to be more controlled, but please provide experimental data. Additionally, instead of PBS, can other physiological fluids be used such as artificial saliva? Please discuss this and provide adequate evidence as this is critical to the novelty of proposed formulations (controlled release profile) Include ethics number for human studies (local ethics and hospital/clinical ethics). Please state number of patients rectruited and power calculations. Please also state how patients were selected and if any exclusion criteria was used and if possible age, sex, comorbidities were included in study design.  Table 2: Please use SD instead of sigma Please include in methods and not results how the nanoparticles were loaded in dental floss. Also provide details of floss/material used as well as loading ability of nanoparticles on dental floss.  Figure 6 statistics are not presented. 

Reviewer 2 Report

Dear Editor,

thank you for the opportunity to review the present paper entitled: "design and evaluation of pH-dependent nanosystems based on cellulose
acetate phthalate, nanoparticles loaded with chlorhexidine for periodontal
treatment .

The revision appers robust and novelty is guaranteed. This research provides insights into a specific field of research and the present manuscript offers valuable data in the invitro  model.

After revision this Reviewer recommend publication of the present manuscript in its present form.

Yours sincerely

Reviewer 3 Report

The paper entitled "Design and Evaluation of nanosystems based pH dependent cellulose acetate phthalate, chlorhexidine loaded nanoparticles for periodontal treatment" develops nanospheres (NSphs) and nanocapsules (NCs) loaded with chlorhexidine base for the treatment of periodontal disease. NCs using the emulsion-diffusion technique including eugenol oil during the manufacturing process. These NCs are compared with NSphs which are produced by the same process without including eugenol oil. Both nanosystems are physicochemical characterization according, size, zeta-potential, to process efficiency, encapsulation efficiency, SEM and DSC studies. Release studies showed a sustained assignment of NCs compared to NSphs. Finally, clinical studies in patients with periodontitis showed the greater decrease of dentobacterial plaque for treatments with NCs compared to mouthwash.

The work is well written and uses appropriate techniques to characterize these nanosystems. The results in vitro and in vivo indicate that these NCs may be a promising treatment for periodontal disease.

This paper could be improved by including the following points.

-lines 325-332. Process efficiency is well explained for NCs but the authors must include the decrease observed in the process efficiency NSphs and include bibliography explaining this decrease.

-line 332. Comment on the higher entrapment efficiency values of NSps compared to NCs (with their significant difference), indicating other references that justify these differences.

-line 337. I think that to clarify the text regarding eugenol oil it must be rewritten. The authors could include after "determined by the HPLC method described above" (lines 336-337) the paragraph "the loading capacities for the batches of NCs were 2.72% ..." (lies 346-349), and end with the reference of "Ying Shao et al., reported an EE of eugenol …” (lines 337-340).

-Line 347. Regarding the increase in the proportion of eugenol oil observed by increasing the amount of CHX, it could be attributed to the fact that increasing the amount of CHX within NCs decreases the evaporation of eugenol.

-DSC studios. I consider that this section can be improved by including the physical mixture in the text. For example, in line 355 after pure CHX include. PM showed a broad peak between the melting temperatures of eugenol oil and CHX corresponding to the interaction of two substances CHX. The decrease in the endothermic peak of CHX in this PM could be attributed to a dilution effect. In addition, Figure 3e showed slight endothermic changes that indicate the presence of PVAL and CAP.

-Line 368. It should be noted that the amount of 10 mg of NCs and NSphs is used for the elaboration of the formulations in these studies. This 10 mg should also be included in the NCs and NSphs of the legend of Fig. 4.

-line 446. O'Leary Index must be cited bibliography of this index or comment on how it measures % reduction.

-line 447. The NCs showed % reduction between 76% (week 1) to 26% (week 3). These important reductions could be commented and compared with other treatments.

-line 448. The authors indicate a 91-67.5% reduction using a commercial product (mounthwash). These results should be commented and discussed with other previous studies.

Round 2

Reviewer 1 Report

Thank you for revising the manuscript
